# Crustal-scale structures and tectonic domains of the Kheis Tectonic Province in South Africa from multimethod seismic analysis

Michael Westgate<sup>1,2</sup>, Musa S. D. Manzi<sup>2</sup>, Alireza Malehmir<sup>1</sup>, Ian James<sup>2,3</sup>, Christian Schiffer<sup>1</sup>

<sup>1</sup>Department of Earth Sciences, Uppsala University, Uppsala, 75236, Sweden

<sup>2</sup>School of Geosciences, University of the Witwatersrand, Johannesburg, South Africa, 2017, South Africa

<sup>3</sup>BHP Billiton Limited, Melbourne, 3000, Australia

Correspondence to: Michael Westgate (Michael.Westgate@geo.uu.se)

**Abstract.** The Kheis Tectonic Province of southern Africa represents a key, yet under-constrained, component in the tectonic history of the Kaapvaal Craton and its surrounding terranes. The complex geological framework is masked by extensive sedimentary cover and limited outcrop, making geophysical investigations essential. In this study, we present an integrated seismic analysis using the combination of a legacy deep reflection data (GS-02), a shallower reflection profile (KBF-01), teleseismic receiver functions, and refraction tomography to refine the crustal architecture and major tectonic boundaries across the region. Prestack time migration of the GS-02 profile reveals significant improvement in reflector clarity, enabling the identification of thrust faults, fold structures, and previously unresolved reflective packages. Refraction tomography constrains the thickness of the Kalahari Group cover, averaging ~250 m, while receiver function analysis at three broadband seismic stations yields new Moho depth estimates ranging from 32 to 46 km and delineates crustal stratification. Our interpretation supports a model in which the Kaapvaal Craton is underthrusting westward beneath the Kheis Province, with partial crustal imbrication. We find no strong seismic evidence for the Dabep Thrust as a major tectonic boundary, aligning with recent challenges to its significance. In contrast, the Blackridge Thrust and the Kalahari Line show coherent seismic and geophysical expression, supporting their role as first-order structures. Additionally, we image a deeply buried, high-reflectivity zone in the west, suggestive of a possible plutonic body or relict basin structure. This multi-method seismic investigation advances the understanding of the crustal-scale structure and tectonic evolution of the Kheis Province, providing new constraints for regional tectonic models and highlighting the value of reinterpreting legacy seismic data with modern techniques.

#### 1 Introduction

Seismic methods have become indispensable tools in the analysis of crustal-scale structures and the delineation of tectonic boundaries, providing detailed insights into the subsurface architecture of the Earth, particularly where surface outcrops are often obscured by younger sedimentary cover. Seismic refraction surveys, while commonly used for near-surface investigations into seismic velocities (with many literature examples, the reader is referred to the following select few: Yordkayhum et al., 2007; Clowes et al., 2010; Gomo et al., 2024; Kucinskaite et al., 2025), have also been widely used for characterizing deep velocity features and even detecting the Mohorovičić (Moho) discontinuity (e.g. Zelt et al., 2003; Buntin

et al., 2021; Soares et al., 2018). Seismic reflection profiles have been used for deep imaging of important and complex tectonic zones that are often associated with orogenic fold belts, terranes, craton margins, and important mineral-forming regions (e.g. Tinker et al., 2002; Daly et al., 2014; Juhlin et al., 2016; Westgate et al., 2020). Finally, teleseismic methods such as receiver function analysis is often used to constrain geometries and depths of regional crustal composites and thickness (e.g. Schiffer et al., 2024). Studies combining various seismic and other geophysical methods have the strength of studying the subsurface using a set of methods with various sensitivities (i.e. Vp, Vs, vertical, horizontal velocity variations) and resolutions, allowing to exploit their respective strengths, and bridge weaknesses or data gaps (i.e. Schiffer et al., 2021; Dentith et al., 2018).

Fault zones, crustal discontinuities and other features that are targeted during such investigations can be difficult to resolve or even detect due to the complex seismic wavefield often associated with such structurally complex and hard rock environments. Resolution of these challenges can be achieved through, for example, integration of complimentary geological and geophysical datasets (e.g. Westgate et al., 2022). Migration of seismic reflection data also plays a significant role in hard-rock seismic analysis by collapsing diffraction signals associated with structural discontinuities, and placing signals from dipping reflectors in their correct subsurface location. It is because of the accurate imaging capabilities of migration, given a sufficiently accurate velocity model, that much research has been conducted into improving migration algorithms and using them in structurally complex geological settings (Buske et al., 2015; Ding and Malehmir, 2021). With the evolution of recent technology, more sophisticated migration algorithms are computationally affordable and with sufficient data quality, prestack migration typically offers the best results (e.g. Sihoyiya et al., 2022). Additionally, modern and more advanced processing algorithms have been used to successfully extract novel and valuable information from legacy data (Malehmir et al., 2019; Manzi et al., 2019; Juhlin et al., 2025).



The Kaapvaal Craton, one of the oldest and most stable Archean cratons, forms the geological core of southern Africa and is renowned for its well-preserved lithospheric architecture (Figure 1). In South Africa, the craton is bounded in the south and east by the Namaqua-Natal Metamorphic Province (NNMP), and in the west by the Kheis and Kaaien Provinces (Figure 1). The region comprising the latter provinces and the implicated boundaries between and around them have been the subject of several studies (e.g. Cornell et al., 2006; Van Niekerk and Beukes, 2019 and references therein). This is especially true in our area of interest, demarcated by the white box in Figure 1. Here, consensus in the literature regarding the structure, composition, and extent of the Kheis and Kaaien provinces, as well as their borders with the NNMP and Kaapvaal Craton, is not well consolidated. This is due, primarily, to the extensive sedimentary cover of the Kalahari Group that yields sparse outcropping of underlying strata. Models of the tectonic layout of the area are based primarily on geophysical data, but the complexity of subsurface structures makes geophysical interpretation, especially of data from magnetic and gravity surveys, challenging and non-definitive. The geophysical data available in the study area include magnetic and gravity maps, as well as a 200-km-long reflection seismic survey (GS-02, record length 16 s) that was acquired and processed in the early 1990s and was documented by Stettler et al. (1998, 1999) together with magnetotelluric single-station soundings in a time of underdeveloped processing

algorithms and outdated geological research in the area. Adjoining this profile to the east is another smaller seismic profile (KBF-01, record length 6 s), which was recently reprocessed and presented by Westgate et al. (2020). Furthermore, the region has been covered by broadband seismic stations part of the temporary SASEK network (XA, <a href="https://doi.org/10.7914/SN/XA\_1997">https://doi.org/10.7914/SN/XA\_1997</a>), as well as the permanent South African National Seismograph Network (AF, <a href="https://doi.org/10.7914/SN/AF">https://doi.org/10.7914/SN/AF</a>).

Figure 1: Map showing the major tectonic provinces of southern Africa and the location of our study region, along with the seismic profiles overlaid in unicolour white (Figure 2 shows independent profiles). *RP* Rehoboth Province; *KP* Kheis Province (here inclusive of the Kaaien Terrane); *KC*; Kaapvaal Craton; *NNMP* Namaqua-Natal Metamorphic Province; *GB* Gariep Belt; *CFB* Cape Fold Belt.


In this study, we use a combination of different seismic methods, incorporating the deep seismic profile (GS-02) for both refraction and reflection data analysis with the shallow seismic reflection profile (KBF-01), and receiver functions from nearby broadband seismic stations, to retrieve novel information about the tectonic structures within and around the Kheis Province. Interpretation of these seismic datasets in conjunction with the magnetic and gravity maps, as well as recent literature on the geology of the area, provides an additional basis of evaluation of the currently competing tectonic models.

Our goals are as follows. Firstly, improving quality and imaging capabilities of the reflection profile using updated processing flows and a Kirchhoff prestack time migration as a core step. Additionally, a reappraisal of regional broadband seismic stations using receiver function analysis to constrain Moho depths and crustal stratification. Secondly, we aim to constrain the depth to the Kheis basement of the Kalahari sands along the seismic profile using refraction tomography, thus providing a thickness profile of the Kalahari cover. Thirdly, we characterize and evaluate major provincial boundaries between tectonic units within the study area in Figure 1. This includes an evaluation of recent proposals by Van Niekerk and Beukes (2019) that challenge, from a geophysical perspective, the existence of the Dabep Thrust as a major tectonic boundary between the Kheis and Kaaien terranes as adopted in pre-existing models, and the interpretation of two additional thrust faults and a series of fold structures

as proposed by the same authors within the Kheis Province. Finally, we outline any novel features obtained from reprocessing the GS-02 reflection seismic profile and consequent bearing on original interpretation by Stettler et al. (1998, 1999).

Through reprocessing and reinterpretation of the available seismic data using modern techniques, we aim to consolidate the various datasets present in the site location with the aim of contributing to an overall solution to these existing questions.

# 2 Regional Geology and Tectonic Layout


The study site is located across four tectonic provinces that decrease in age from east to west. These provinces, as well as their structural boundaries, are delineated clearly by the aeromagnetic map shown in Figure 2. Additional datasets available in the region include the gravity and the seismic datasets: the 200-km-long GS-02 seismic profile, the adjoining KBF-01 seismic profile and the seismic stations UPI, SA22 and SA23.

Figure 2: Geophysical and geological maps of the study area, including (a) aeromagnetic, (b) elevation, (c) gravity and (d) outcrop 100 geology. Major tectonic boundaries are labeled on the aeromagnetic map. The two profiles used in the study, GS-02 and KBF-01, are plotted in black and red, respectively. The straight portion of the GS-02 profile that is highlighted in white demarcates the extent of the tomography. White circles with labels mark the broadband seismic stations. Inferred thrust traces TA ("Thrust A": Van Niekerk and Beukes, 2019), TB ("Thrust B"; Van Niekerk and Beukes, 2019), and DT (Dabep Thrust; Moen, 1999) are also plotted. Major terranes are labeled as: NMMP - Namaqua-Natal Metamorphic; Kh/Ka - (Kheis / Kaaien provinces); KC - Kaapvaal Craton. Coordinate reference system used: Web Mercator Projection.


The study area and the tectonic provinces contained therein have been studied for a long time, with major contributions by, for example, Thomas et al. (1994), Stettler (1999), Moen (1999), Tinker et al. (2002), Cornell et al. (2006), Moen (2006), Moen and Armstrong (2008), and Van Niekerk and Beukes (2019). In general, the tectonostratigraphy of the area consists of mostly

Paleoproterozoic crust that has been reworked and overprinted by the polyphase Namaquan Orogeny during plate collision and the assembly of Rodinia in the late Mesoproterozoic. Details pertaining to ages, lithostratigraphy, and the tectonic setting have been a source of controversy in the literature, due mainly to obscured outcrop and limited constraining data (e.g. well data and geophysical coverage). Additionally, there have been multiple proposals for the tectonic model, with no unanimous candidate. In a recent a study, Büttner (2020) challenged one of the more favorable collisional tectonic models of this region, citing a lack of evidence of subduction-related metamorphism such as blueschists, eclogites, or low-temperature/high-pressure indicators. In this study, we seek not to address the validity of any one proposed tectonic model, but rather present updated geophysical evidence relating to some of the main structural boundaries from the contrasting studies. Other data, such as surface geology and elevation aid in constraining our interpretations. Publicly available deep borehole data is limited in the area, with none recorded over the Kheis Province.






110

115

To the east of our study site, where the geology is better constrained by outcrop and geophysical data, is the western boundary of the Archean Kaapavaal Craton. Here, the craton is characterized by a thick sequence of westward-dipping supracrustal metasediments (Tinker et al., 2002; Westgate et al., 2020). The uppermost of these layers belong to the Ghaap and Postmasburg Groups of the Transvaal Supergroup in the Griqualand West area, which contain banded iron formations that are responsible for the striking arcuate magnetic feature that runs northwards until displacement of these units by the Moshaweng Fault (Figure 2a; Westgate et al., 2021). In their interpretation of the GS-02 seismic profile, Stettler et al. (1999) interpreted some of the upper units of the associated reflection package as ophiolites that were obducted onto the Kaapvaal Craton during the formation of the Kheis Province. The primary justification for their interpretation comes from the excess mass that was needed in their density model that attempted to account for the large Bouguer anomaly near the eastern end of the seismic traverse (as seen in Figure 2C). The authors also noted the substantial thickness in the corresponding reflection package, which was used to justify the ophiolite interpretation. In a later study, Tinker et al. (2002) examined another, shallower (6 s long) seismic profile that adjoins the GS-02 profile on its eastern end. The consequent interpretation of this profile, labelled KBF-01 (Figure 2), strongly contrasted the model proposed by Stettler et al. (1999), with no reference to ophiolites and a stratigraphy that contained exclusively Kaapavaal supracrustal rocks. These authors also noted a series of reflections that do not outcrop and constitute the base of the supracrustal rocks, which they labelled UA, UB and UC (as adopted in this study), likely linked to the surplus reflectors mentioned by Stettler et al. (1999). The same profile KBF-01 was later reprocessed by Westgate et al. (2020) for iron-ore-targeted imaging and is reproduced in the results of this study. Interpretation of the profile was aided by the boreholes in the area (Figure 2; Westgate et al., 2021; Westgate et al., 2022). An open question that is necessary in understanding the tectonics of this area is whether the seismic horizons of the Kaapvaal Craton in profile GS-02 can be correlated with those of KBF-01.

To the west and overlying the dipping units of the cratonic crust are rocks from the Olifantshoek Supergroup, comprising mostly meta-arenites, that were thrust onto the Kaapvaal Craton. The Kheis Province is hence delineated; an east-verging fold

and thrust belt whose north-south fabric is observable in the region's magnetic map (Figure 2a). Originally termed the Kheis Orogeny with an age of 1.8 Ga (Cornell et al., 1998), the fold-and-thrust event, albeit still distinct, has now been taken as the earliest stages of the ~1.2 Ga Namaqua-Natal orogeny, since the zircon dating study by Moen and Armstrong (2008). The eastern boundary of the Kheis Province with the Kaapvaal Craton, also termed the Kheis Front, is coincident with the Blackridge Thrust fault (Figure 2; Van Niekerk and Beukes, 2019). The western limit of the Kheis Province has traditionally been placed at the Dabep Thrust, where it is in faulted contact with the western Kaaien Terrane (Figure 2; Moen, 1999). The latter is characterized by metaquartzites and schists that exhibit a transition from the lower-grade Kaapvaal and Kheis rocks to those of high-grade amphibolite-facies metamorphism in the Areachap Terrane to the west and is interpreted as the eastern foreland of the Namaquan sector of the NNMP (Cornell et al., 2006; Pettersson, 2007). In the west, the Kaaien Terrane is bounded by the dextral Brakbosch-Trooilapspan shear zone (Figure 2).

145

150

170

Tectonic, stratigraphic, and age attributes of the Kheis and Kaajen domains have been significantly debated in the literature and, most recently, Van Niekerk and Beukes (2019) have proposed merging them into a singular "Kheis Terrane", which hosts the proposed restructured "Keis Supergroup". Their primary justification for such changes is the purported lack of evidence for the Dabep Thrust's status as a major tectonic boundary, and the authors support their claims with outcrop and satellite studies and, more pertinent to this study, geophysical data. Specifically, the trace of the thrust originally proposed by Moen 160 (1999), and generally accepted in the literature, crosscuts the regional magnetic fabric (Figure 2a), and is not consistent with the associated tectonic structures, a claim that is also supported by Corner and Durrheim (2018). The newly proposed Kheis Terrane is suggested to be bound in the east, as in earlier interpretations, by the Blackridge Thrust, and in the west by the Kalahari Line (Figure 2), a significant magnetic feature that has been interpreted as a major suture along the western boundary of the Kaapvaal Craton that separates eastern shallow basement from deeper western basement in both South Africa and 165 Botswana (Corner and Durrheim, 2018). In addition to the refutation by Van Niekerk and Beukes (2019) of the Daber Thrust's presence in the study area, these authors inferred two separate thrust faults in the area, one near the Skurweberg mountains labelled "TA" in Figure 2, and one further east towards Olifantshoek, labelled "TB".

Moving further west, the region demarcated by the Brakbosch-Trooilapspan shear zone is accompanied by a series of subparallel, NW-plunging fold structures that were identified by Van Niekerk and Beukes (2019) as the Orange River syncline, the Gariep anticline and syncline, and the Koras anticline based on the aeromagnetic data (Figure 2). These are attributed to crustal shortening of the rocks contained in the Kheis and Kaaien domains. The shear zone, which separates the eastern Kaaien Terrane from the western Areachap Terrane is suggested to have both lateral and vertical components of uncertain displacement (Moen and Armstrong, 2008). The Areachap Terrane, distinct from the Areachap Group and comprising mostly granitoids, has been interpreted as the modern constituents of a Namaquan-age volcanic island arc in the subduction model, later intruded by the Kheimos Suite (Figure 1; Cornell et al., 2006; Van Niekerk and Beukes, 2019). In the region separating the Brakbosch

Fault and the Trooilapspan Shear Zone lies an enigmatic zone that was suggested by Van Niekerk and Beukes (2019) as back-thrusted units of the Brulpan and Wilgenhoutsdrif Groups (Figures 2).

Covering most of the Paleoproterozoic rocks in the study area, with occasional outcrops, are the Cenozoic Kalahari Group sands (Figure 2; Haddon and McCarthy, 2005; Matmon et al., 2015). The Kalahari Group isopach map by Haddon (2004), based on pre-existing maps, reports, borehole and geophysical data, suggests the thickness of the Kalahari sediments near our study area to reach up to about 60 m. Outcrops along the seismic profile traverse include (Figure 2c): various Areachap-related metasediments forming the Kaaien Hills near the western end; Brulpan Group quartzites that constitute the Skurweberg Mountains near the profile's midpoint, and; Transvaal and Olifantshoek unit outcrops close to the Langeberg Mountains and within the Maremane Dome to the east. The Kalahari sands occupy most of the space proposed by Van Niekerk and Beukes (2019) to be the domain of the Kheis Terrane, that is, the area bounded by the Brakbosch Thrust in the east and the Kalahari Line in the west (Figure 2). Coincidentally, it is over this domain that profile GS-02 is straightest, providing favorable conditions to gauge the Kheisan cover thickness using a first-arrival, ray-based tomography from the seismic profile.





Crustal thicknesses and depth-to-Moho across the Kaapyaal Craton have been obtained primarily through teleseismic receiver function (RF) analysis of events recorded at seismic monitoring stations throughout southern Africa (e.g. Kgaswane et al., 2009; Youssof et al., 2013; Baranov et al., 2023) or ambient noise surface wave tomography (Yang et al., 2008). Most of these stations were set up during the South African Seismic Experiment in the 1990s (SASEK; Carlson et al., 1996), while some are part of the South African National Seismograph Network (SANSN). Moho depth beneath the Kaapvaal Craton ranges from 35 to 45 km, with a relatively sharp and strong velocity contrast (Corner and Durrheim, 2018). Models of crustal thickness within our region of study vary quite significantly (Baranov et al., 2023; Youssof et al., 2013 and references therein), but generally show a thickening of the Kaapvaal Craton near its border with the Kheis Province, followed by a zone of thinning to the west beneath the Kheis Province, and then thickening beneath the NNMP (Kgaswane et al., 2009; Baranov and Bobrov, 2018). For our study, we use three stations that run roughly parallel to the seismic profile to compute receiver functions and invert for seismic velocities: the SANSN UPI station to the west, and the SASEK stations SA22 and SA23 stations near the central and eastern portions, respectively (Figure 2). According to Kgaswane et al. (2009), crustal thicknesses for stations UPI, SA22 and SA23 are calculated at 40, 35 and 40 km, respectively. These results are generally supported by authors of other RF studies, such as Nair et al., 2006 (40.4 km for SA23) and Nguuri et al. (2001) (35 km for SA22 and 44 km for SA23), while Youssof et al., (2013) obtained a thicker crust for SA22 (48 km), but similar crustal thickness for SA23 (41.5 km). The tomography study of Yang et al. (2008) confirms a ~30 km thick crust beneath SA22 and ~40 km beneath SA23.

# 3 Data and Methodology

# 3.1 Seismic reflection profiles

To address the modern geological questions and recent hypotheses about the Kheis Province, we reanalyzed the GS-02 seismic reflection profile, a legacy dataset from the 1990s. Using updated reflection and refraction techniques, including advanced prestack migration and velocity modeling, we reprocess this dataset to improve imaging. Complementary receiver function studies from regional broadband seismic stations further refine our understanding of crustal structure and Moho depths, providing the necessary framework to achieve the goals of this study.

The GS-02 seismic reflection profile was acquired in 1991 using a split-spread roll-along geometry to ensure consistent subsurface coverage. The survey employed a receiver spacing of 50 m and a shot spacing of 150 m. Four seismic vibrator trucks served as the energy source, generating sweeps ranging from 8 to 64 Hz over 16 s. Data were recorded at a 2 ms sampling interval, with a maximum offset of 4775 m, and a total recording time of 32 s. After correlation with the sweep, 16 s of the data were used for processing and imaging.




The reprocessing of the GS-02 seismic reflection profile was designed to maximize the imaging of crustal structures by applying a modern workflow, detailed in Table 1. Key processing steps included defining geometry, refraction static corrections, amplitude recovery, noise suppression using 1D and 2D filters, velocity analysis, residual static corrections, and, finally, prestack time migration. The lattermost, which was performed using a 2D Kirchhoff migration algorithm, yielded the most significant improvement in the dataset. To ensure optimal results, the velocity model was constructed sequentially and iteratively, starting from the shallow subsurface and progressively building the velocity model into deeper layers. Figure 3 illustrates a comparison of different portions of the reflection profile without and with the prestack migration. Panels a and d highlight fold and thrust features that are only resolved after migration, panels b and e show the correct positioning by the migration of dipping reflectors, as well as the enhancement of more subtle parallel reflectors, and panels c and f demonstrate how the migration has revealed a strong package of complex reflection signals at depth that has been constructed from deep diffraction signals. The various features observed in Figure 3 are typical of seismic data collected over fold and thrust belts, and hard rock environments, demonstrating the necessity for effective migration in such settings.

| A. Pre-processing                    | B. Pre-stack processing           | C. Migration               |
|--------------------------------------|-----------------------------------|----------------------------|
| 1. Trace editing                     | 1. Initial static corrections     | 1. Pre-stack preparation   |
| Kill noisy traces and check polarity | Floating datum statics            | Time-varying bandpass,     |
|                                      | Repl. Vel. 2500 m.s <sup>-1</sup> | Amplitude gain: spherical  |
| 2.Geometry                           | Refraction statics                | divergence                 |
| CMP binning and corrections          |                                   | Mute first arrivals        |
| Nominal bin spacing: 25 m            | 2. Wiener deconvolution           |                            |
|                                      | Predictive deconvolution          | 2. Pre-stack migration     |
|                                      | Gap: 4 ms                         | Kirchoff 2D time migration |

Filter length: 92 ms

Bandpass 8-65 Hz

3. Frequency filtering

# 4. Ground-roll removal

Radial trace filtering: low pass 8 Hz subtracted from data

# 5. Offset regularization

New offset intervals: 12.5 m

# 6. Residual statics

Surface-consistent

Aperture: 8.5 km

Maximum migration dip angle: 70°

### 4. Mute and stack

Post migration mute Stacking

# 5. Post-migration processing

Time-variant bandpass filter Semblance filter

Time-to-depth conversion

Table 1: Steps used for reprocessing the GS-02 survey data.


Figure 3: Three windows of the GS-02 stack section without (a-c), and with (d-f) prestack migration, showing a significant enhancement in imaging capabilities at varying depths. Ubiquitous diffraction patterns reveal detailed structure through migration of these signals.

# 240 3.2 First-arrival tomography

For tomographic analysis, first breaks were picked at full offset across the profile and extracted from the portion highlighted in Figure 2. An example shot gather is shown in Figure 4a, alongside the global selected picks plotted the offset-time domain (Figure 4c). Most shot gathers are characterized by first arrivals that grouped into two apparent velocity trends: one at 2900 m.s<sup>-1</sup> at the near offsets and one at 5500 m.s<sup>-1</sup> at further offsets.

Figure 4: (a) Example shot gather with first arrival picks overlaid in blue, and (b) global picks plotted in offset-time domain. Both panels have two velocity slopes plotted, corresponding to two dominant arrival velocities.

The tomographic inversion of the first breaks was conducted by inverse modelling the seismic velocity field from the traveltimes of first arrivals that were picked from the shot gathers. Traveltimes were forward-modeled using a finite-difference method that approximates the Eikonal ray equation, and the inversion method made use of an iterative conjugate-gradient least-squares method (as detailed in Benz et al., 1996; Tryggvason et al., 2002; Rodríguez-Tablante et al., 2006).

# 3.1 Teleseismic data analysis





In order to assist the deep seismic reflection data with independent constraints on crustal stratification and Moho depth, we apply joint inversion of RFs and apparent S-wave velocities (Vsapp) at three stations of the Africa Array, which are nearest to the seismic profiles (stations SA22 and SA 23 of the SASEK network, network code XA, Silver, 1997, and station UPI of the South African National Seismograph Network, network code SQ. RFs and Vsapp are derived from teleseismic recordings at three-component broadband seismometers.

RFs yield an approximation of the Earth's seismic impulse response beneath a station by deconvolving the incoming P-wavefield from teleseismic earthquakes with the P-to-S (Ps) converted wavefield (Langston, 1977; Vinnik, 1977). The deconvolution process mitigates the influence of source characteristics, propagation path effects, and instrumental response,

thereby isolating Ps conversions as discrete pulses. An RF encompasses the direct P-wave arrival at 0 seconds, primary Ps conversions originating from each subsurface velocity discontinuity, and additional conversions arising from free-surface multiples (e.g., Langston, 1977). The seismograms were transformed from the initial Z-N-E (vertical-north-east) coordinate system into R-T-Z (radial-transverse-vertical) components and bandpass-filtered between 0.03 and 8 Hz. A frequency-domain deconvolution, employing a water-level stabilisation parameter of 0.01, was then performed. Subsequently, RF waveforms underwent both automated and manual quality assessments, with those displaying excessive noise, implausible P-wave amplitudes, or prominent long-wavelength artefacts being excluded.

The Vsapp parameter characterizes the polarization of incoming teleseismic P-waves across a range of frequencies, providing insights into the S-wave velocity structure beneath a seismic station. This is determined from the ratio of radial (R) to vertical (Z) receiver functions at zero lag time (Svenningsen & Jacobsen, 2007). The resulting velocities are apparent, representing an integrated measure of the vertical structure sampled by the dominant wavelength of the incident P-wave. We generate Vsapp values at incrementally longer periods (T), which probe greater depths, following the approach outlined by Svenningsen & Jacobsen (2007). These periods comprise 51 logarithmically spaced values ranging from 1 to 25 seconds.

For the inverse modelling, we apply the inversion algorithm by Schiffer et al. (2023) that jointly inverts for both RFs and 275 Vsapp through combination of linearized iterative least squares (LLSQ) inversion (e.g. Tarantola & Valette, 1982) and a random model search scheme. For each station, we perform 1000 LLSQ runs with random starting models, saving the last 10 iterations to form a posterior model distribution of 10,000 models. The starting models have 6-18 layers, the velocities are constrained between 1 and 5 km.s<sup>-1</sup>, and monotonously increasing with depth, the maximum of which is constrained to 80 km. Each inversion runs for 15-50 iterations. Stacked RFs are modelled from -1 to 25 s delay time, and Vsapp curves are computed 280 for periods from 1 to 25 s. A priori data errors, based on covariance matrices, weight the datasets. Laver parameters are defined by Vs and delay times to reduce non-linearity (Jacobsen & Svenningsen, 2008). Vp/Vs ratios are determined using lithologybased lookup tables (Christensen, 1996), and densities are constrained by Vp (Christensen & Mooney, 1995), During inversion, Vs and delay times can freely change, except for the deepest velocity, which is constrained with a prior model error of 0.2 km.s<sup>-1</sup>. Each LLSQ inversion provides data, model, and roughness errors (Qd, Qm, and Qr), with total error (Q) as their sum. 285 The inversion stops when the total error changes by less than 0.1% for two consecutive iterations or after 50 iterations. The final model is selected from the posterior population based on maximum density, with posterior model error defined by the standard deviation of the population.

#### 4 Results and Interpretation

#### 4.1 Reprocessed reflection profile

The reprocessed GS-02 profile is shown in its entirety in Figure 5 (the amplitude envelope is plotted for display purposes). A time-to-depth conversion was performed using a velocity model obtained from converting the RMS velocity model, which itself was constructed from constant velocity analysis of the reflection signals, to interval velocities and applying a series of

smoothing filters. The maximum depth after time-to-depth conversion is approximately 40 km, with obvious reflections observed down to 38 km depth. In general, the section exhibits a series of strong and mostly clear dipping reflections in the eastern portion of the profile, followed by chaotic reflectivity moving to the west, with multiple truncated and localized reflections at varying depths. The western half of the profile also has various localized reflectors, including a large, deep concave-up reflection package. The far western end is mostly transparent.

305

310

Figure 5: The amplitude envelope, or instantaneous amplitude, of the reprocessed GS-02 seismic section plotted in 3D space. Boxes indicate portions of the profile that are enlarged in subsequent images. Plotted on top of the seismic section is the geological map as in Figure 2.

Before reviewing the details of the final GS-02 section, we compare the reprocessed results with the original results that were presented by Stettler et al. (1998, 1999). This is presented in Figure 6, where an eastern portion of the two sections is highlighted. A striking difference is observed in signal clarity and coherence. It was found during processing that the most substantial improvements in the reprocessed data were attributed to the prestack time migration. This contrasts strongly with the post-stack migration that was likely used in the original processing. To further improve reflection coherence and structural mapping, a semblance filter was applied after the migration (see insets on Figure 6). Structural features such as thrust units are more clearly resolved, as shown in the zoom window of Figure 6, and layer boundaries in the dipping units are more easily defined especially where they contact the thrust layers.

Figure 6: A comparison window of the GS-02 profile from (a) the 1998 migrated section, and (b) the reprocessed and pre-stack migrated section, with inset windows showing a portion of the reprocessed section with and without semblance filtering.

The eastern portion of the profile is shown alongside the KBF-01 profile from Westgate et al. (2020) in Figure 7. Seismic horizons correlate well across the two profiles and structures can be tracked effectively from the more-documented KBF-01 profile (Tinker et al., 2002; Westgate et al., 2020). This portion of the reprocessed GS-02 profile is characterized by westward-dipping horizons, with an average dip of 25°, that are a clear continuation of the supracrustal horizons identified in KBF-01, where they flatten out towards the east, where they outcrop (Figure 7). Based on the interpretation by Tinker et al. (2002), these seismic horizons represent units from the Ghaap and Postmasburg Groups of the Transvaal Supergroup, which overlies the sediments and lavas of the Ventersdorp Supergroup. The total thickness of the reflective package, from the base of the Ventersdorp sediments to the top of the Postmasburg Group, averages 6 km in the east where they are flat, and 9 km where they dip towards the west. The boundary of their western extent is obscured by a lack of reflectivity, but they appear to pinch out at depths of 20 to 30 km. Beneath the Ventersdorp sediments are three layers of distinct reflectivity, marked  $U_A$ ,  $U_B$ , and  $U_C$  in Figure 7 after Tinker et al. (2002), that do not outcrop and pinchout beneath the Ventersdorp sediments in the east. These units attain a maximum collective thickness of 8 km where the two seismic profiles intersect.

There are some localized reflective zones beneath  $U_A$  in both seismic profiles; in the east, there appear to be three dipping zones of concentrated but incoherent reflectivity, which extend beyond the record length of the KBF-01 profile. Extrapolation of these zones to the west leads to an intersection with another localized horizontal reflection at a depth around 26 km in GS-02, suggesting a possible link. Further west, at depths of about 33 km, is another pocket of localized reflections with a gentle westward dip (see zoom box in Figure 7).



The Blackridge Thrust is identified in the seismic section as a reflection of varying amplitude that has an average dip of 28° and extends to the surface at the points that corresponds with surficial mapping. It underlies a cluster of chaotic reflections that exhibit multiple lenticular features accompanied by semi-continuous reflections of conflicting dips. These are likely caused by thrust structures within the Olifantshoek Supergroup. Above these is a region of transparency, followed by a zone of arbitrary reflectivity and reflections with varying dips and curvature.

Figure 7: Eastern portion of the GS-02 line (a) as outlined in Figure 5, with CMP bins labelled and starting at 10000. Seismic profile KBF-01 is also plotted, with CMP labels retracted for simplicity. Projection of seismic station SA23 coordinates onto seismic line is also illustrated with pink label. Major features are highlighted and interpreted in (b).




The central portion of GS-02 is shown in Figure 8. Moving westward away from the Blackridge Thrust, a transparent zone or layer is seen, adjacent to the complex thrust units, that conforms with the orientation of the eastern dipping units, followed by a region of sparse sub horizontal reflections that exhibit mild fold and thrust features between CMPs 12500 and 13500. Beneath this region, at depths between 15 and 26 km, and just off the deepest extent of the interpreted Kaapvaal Craton reflectors, lie several parallel reflections that curve upward to the east. While not clearly bounded, the internal reflectivity of these sets bears strong similarities to the upper layers of the Kaapvaal reflectors, suggesting a potential connection. Either these packages belong to the Kaapvaal Craton and have been stretched out, or they resemble a portion of exotic Kheis basement. West of CMP 13500 the general paradigm of reflector properties shifts to eastward-dipping (averaging 27°), localized reflections with mild folds. These signals appear to emerge from a relatively low-signal zone and are abruptly truncated up-dip by a sub horizontal reflection above, whose amplitude tapers off, and a transparent column adjacently west. The sub horizontal reflection is accompanied by more lenticular signals that imply thrust faulting. The fault can be tracked to the surface via a primary phase with a low amplitude and a 22° westward dip. These could be tied to backthrusts of post-Olifantshoek units within the Kheis Province. The transparent column lies directly to the west, between CMPs 14800 and 15000, and truncates reflectors on both sides. The lateral position of this column correlates directly with where the Kalahari Line intersects the GS-02 profile in its southernmost extent. Given the width and depth extent of this column, the wavelength of the Kalahari Line anomaly appears relatively short in comparison and is thus likely the result of a much narrower and potentially shallow feature within the column. However, the anomaly is still interpreted to demarcate a major boundary as suggested by Corner and Durrheim (2018) and Van Niekerk and Beukes (2019), and the seismic transparency could be explained by steep reflection planes.

Figure 8: Central portion of the GS-02 line, outlined in Figure 5, with significant surface features in the literature and the magnetic anomaly along the profile (a). Projection of seismic station SA22 coordinates onto seismic line is also illustrated with pink label. Major features are highlighted and interpreted in (b).

Figure 9 shows the western portion of the profile. Starting from the east, a series of concave-up reflections is observed directly west of the Kalahari Line transparent zone, between CMPs 15000 and 15500, and at depths around 10 km. This is located directly beneath the intersection of the seismic profile with the fold trace of the plunging Orange River Syncline (Van Niekerk and Beukes, 2019). Directly to the west, centered on CMP 15550 at 4 km depth, is a set of reflections that exhibit alternating up-down concavity, interpreted as the Gariep Anticline/Syncline pair. Accompanying these signals is a subtle basal reflection that has an eastward dip of about 20°. The crosscutting nature of this reflection and its orientation are reminiscent of a thrust fault. Extending this reflection to the surface results with it coinciding with a local magnetic high near CMP 16200 (Figure 9a). Further below, at depths of 13 to 20 km is a distinctive series of upward-curved reflections. The amplitudes near the top of these reflections are strong and continuous, and are preceded in depth by a strikingly clear zone with little reflectivity,

indicating a strong acoustic impedance contrast and an absence of any notable reflectors at shallower depths (except within the first 5 km). The reflections within the package group into two sets of crosscutting reflection patterns, one horizontal, and one that curves upward in the western direction. Stettler et al. (1998) noted that a similar and parallel feature is observed in other seismic sections in Botswana, and that little is known about its nature. The package bears similar characteristics to a buried valley, potentially sill-intruded, but explanations as to how such a valley could be buried and preserved at depths of 20 km require more investigations. The upward-concave shape, along with the well-bounded lateral terminations of reflectivity, could alternatively indicate a filled extensional structure, suggesting a more plausible interpretation of a syn-tectonic sedimentary basin or a relict rift basin formed during an extensional phase. If this is true, it explains the sudden truncation of the package flanks as bounding faults. West of this package, from about CMP 16300 eastwards, reflections become sparse and incoherent, impeding the clarity of interpretation. Notable reflections include collection of parallel, east-dipping reflectors between CMPs 16500 and 17000 at depths from 20 km to 27 km, as well as some shallower, isolated packages that all have roughly the same westward dip, and extend from the surface near CMP 16500 to a 15-18 km depth at the western edge of the seismic profile. The geometry of these packages contrast with the forementioned deeper reflectors, suggesting a different stratigraphic or tectonic setting. In terms of surficial observations, this westernmost section of the profile coincides with the Brackbosch-Trooilapspan Shear Zone. In the literature, the Brackbosch and Trooilapspan structures are conventionally assumed to be connected and continuous. However, Van Niekerk and Beukes (2019) allude to a fault seen in both satellite and aeromagnetic data (Figure 2) that departs from the shear zone somewhere along the transition between the Brackbosch and Trooilapspan components, which is taken as a northern extension of the Brackbosch Fault. This fault, which is seen in Figure 2a extends to the profile where it meets the southern "hook" of the Kalahari Line in the magnetic map. This conjunction coincides with the basal surface extension of the isolated, westward-dipping packages at CMP 16500 of Figure 9. Further west near CMPs 17250-17500 is the surface and magnetic expression of the Troolapspan Shear Zone, which coincides to the upper reflections of the same package. This complex geometry could be suggestive of a broad shear zone, with the Brackbosch Fault extension as its base. In terms of the geology, west of this region coincides with outcrops that belong to the Areachap domain (Figure 2d). The incoherent reflectivity could be the result of a dispersive and chaotic wavefield caused by the crystalline Areachap units. Fabric destruction along the steep fault zone may also explain the scarcity of continuous reflections within the region.






Figure 9: Western portion of the GS-02 line, outlined in Figure 5, with significant surface features in the literature and the residual magnetic anomaly along the profile (a). Projection of seismic station UPI coordinates onto seismic line is also illustrated with pink label. Major features are highlighted and interpreted in (b).

# 4.2 Receiver functions and Vsapp


The joint inversion of RFs and Vsapp yield 1D crustal models for the three stations used, namely UPI, SA22 and SA23 from west to east, which are plotted in Figure 10. Included in the plots are rose diagrams of the backazimuth distributions of the events used. For further depth of analysis, the reader is referred to Appendix A2, where selective stacks of the RFs are grouped by into north-south-east-west quadrants. While these models only provide a regional sampling of the crustal structure, due to

their large station offsets, they assist our interpretation of the reflection seismic images, especially in the case of the Moho that is often elusive in reflection seismics. The results from all three stations indicate a clear and rather sharp crust-mantle transition, with the Moho interpreted in the centre of this transition. The interpretations of the crustal models also include: interfaces for high velocity lower crust (HVLC, typically characterized by Vs>4.05 km.s<sup>-1</sup>), which is often interpreted to be a mafic lower crust, or lower crust intruded by mafic magmatics; the mid-crust, i.e. the boundary between upper and lower crust, characterized by a discontinuity or change in velocity gradient at which Vs~3.8 km.s<sup>-1</sup> is exceeded; and two horizons in the uppermost crust, which may be related to the interface between sediments (typically Vs<3.0 km.s<sup>-1</sup>) and crystalline basement and/or metasedimentary layers.

For station UPI (Figure 10a) at the western end of the GS-01 seismic profile, with only a few kilometres distance to the line, the upper and lower crust make up about 50% of the crustal column each with the mid-crustal interface at ~22 km and the Moho at ~40 km depth. HVLC again makes up ~50% of the lower crust, and the crust mantle transition is approximately 2 km thick. There is no indication for thick uppermost (sedimentary) layers. The GS-02 reflection seismic profile is relatively transparent at the position of UPI (Figure 9), except for clear structures in the top 10-15 km. The mid-crust beneath UPI corresponds to strong reflectivity on GS-02 approximately 10 km to the east, which may be associated with the upper-lower crustal interface. The top of the HVLC can be continued to changes in regions of reflectivity to the east. Finally, the estimated Moho (~40 km) is at similar depth, but still slightly shallower, as packages of reflectivity to the east and west in the reflection seismic at ~42-45 km depth. This allows confirming the general Moho architecture in the west of the profile at ~40-45 km depth. The earthquakes used are primarily arriving from the east, west and southwest, however plots of RFs binned in four quadrants (north, south, east, west) show little variation at station UPI with the largest peak at ~5 seconds (Figure A2a)

Station SA22 (Figure 10b), situated ~30 km to the north in the centre of the GS-02 profile but still within the same domain of magnetic facies (Figure 2), yields a considerably shallower Moho at ~32 km depth compared to station UPI, and is furthermore characterized by a ~1-1.5 km thick crust-mantle transition. The upper crust is about twice as thick as the lower crust. The HVLC with ~4 km thickness appears to be thinner than at UPI. Station SA22 is located within a region of complex reflectivity in GS-02 (Figure 8). When projected onto the seismic line, the station is surrounded by east-dipping reflectors in the upper crust and west-dipping reflectors to the east. The mid-crustal interface, top HVLC and the Moho can be roughly related to reflectivity at similar depths in the GS-02 profile. The reflectivity, however, seems to be several kilometers deeper than what is estimated by the RFs, which can be well-explained by the 30 km distance between both datasets. Due to the large offset of SA22 from GS-02, we cannot expect structures to coincide very well. The earthquakes used at station SA22 primarily arrive from southwesterly backazimuths. A closer look at the backazimuthal variation of the RFs shows that the RFs in the western northern and eastern quadrants are similar with a distinct peak at ~2.5 sec and a smaller peak at ~4 sec, while the RFs from the southern quadrant show distinct peaks around 5 sec, potentially indicating a crustal thickness gradient, with thicker crust towards the seismic line (Figure A2b).

The deepest Moho was estimated from station SA23 at ~46 km depth. SA23 (Figure 10c) is located at the eastern end of the composite reflection profile, in the eastern half of the shallow reflection profile KBF-01, and almost exactly colinear. The RF-445 Vsapp inversion yields a ~25 km thick lower crust, compared to modest 20 km thick upper crust that includes a ~13 km thick (meta-)sedimentary package, according to the velocity model. This thick sedimentary package itself may be divided into a ~8 km upper group and a ~5 km lower group, which corresponds well with the upper sedimentary packages of the Kaapvaal craton imaged by the KBF-01 seismic reflection line (Figure 7). The mid-crustal interface and the Moho at SA23 allows reasonable correlation with reflectivity at the eastern edge of GS-02. The earthquakes used for inversion at SA23 predominantly arrive from easterly and southwesterly backazimuths. The RF stacks from various backazimuths are reasonably consistent and show peaks at 2-3 sec, 5 sec and ~7 sec, indicating similar crustal structure in all directions, and the presence of intra-crustal multiple reverberations, likely due to the upper crustal sedimentary package (Figure A2c). The positive peak at 2-3 seconds appears to be arriving later from the west and south compared to the north and east, which may indicate the (south?)-westerly dip of the basement interface, although more detailed modelling is required to confirm that.

Figure 10: Inverted Vs models (left), Vs profiles (centre) and Receiver functions (right) of the seismic stations (a) UPI, (b) SA22, and (c) SA23 along the GS-02 reflection profile. Background colours represent the density of 1D models, as well as corresponding synthetic Vsapp and RFs from the 10.000 model population of the inversion. Solid red lines indicate bounding velocities of crustmantle transition (CMT), and red dotted lines indicate velocities of major boundaries at depth. The polar diagrams show the distribution of backazimuths of the earthquakes used.

# 4.3 First-arrival tomography



The results of the tomography, including both computed ray paths and velocities, are shown in Figure 11. There is an evident refractive boundary around which the ray paths cluster (Figure 11b, black arrows), which coincides with horizons in the reflection section (Figure 11a, black arrows). In the velocity plot (Figure 11c), this boundary corresponds with a sharp increase in velocities. In sum, the three plots in Figure 11 suggest a boundary associated with a strong velocity contrast, likely to be linked to the base of the Kalahari Group, where Phanerozoic sediments are in contact with Paleoproterozoic metasediments. This assertion is also supported by the bimodal distribution of the computed velocities, which cluster around velocity values of 2.9 and 5.5 km.s<sup>-1</sup>, an observation that is consistent with the apparent velocities observed in the picked arrivals (Figure 4). The average depth of the boundary is 250 m, with a generally shallower depth in the west. It attains a maximum thickness of 360 m between the Langeberg and Skurweberg mountains, and a zero thickness at the locations where Proterozoic outcrops have been mapped.

Figure 11: Results of the first-break traveltime tomography, showing (a) the reflection section, (b) the computed raypath count, and (c) the velocity field. The colourbar for (c) is plotted as a histogram in (d), and (e) shows the velocity results overlaid on the geology map from Figure 2 to illustrate outcrops. Black arrows delineate a strong lateral velocity contrast that are coincident with regions of reflectivity.

#### 5 Discussion


#### 5.1 Improvement in overall data quality and interpretability

The reprocessing of the GS-02 seismic reflection profile resulted in enhanced imaging quality, with prestack Kirchhoff migration yielding the most substantial improvements, followed by a subsequent semblance filter. Compared to the original seismic section, the reprocessed data exhibit greater reflector continuity, improved signal coherence, and better resolution of

dipping structures and fault zones (Figure 6). The migration process effectively collapsed diffractions and placed reflectors in their true subsurface positions, resolving ambiguities that were present in the legacy dataset. Notably, previously obscured structural elements, such as thrust faults and deep-seated reflective packages, became more distinguishable after migration (Figure 3).

Despite these improvements, inherent limitations of 2D seismic data remain. The absence of out-of-plane information can introduce uncertainties in reflector positioning and interpretation, particularly in regions with complex three-dimensional setting. Additionally, 2D seismic sections may suffer from artefacts due to lateral velocity variations that are not accounted for in the processing workflow. These limitations necessitate cautious interpretation.

The refraction tomography results provided an independent constraint on shallow subsurface velocities, refining the velocity model used for prestack migration and improving depth conversions in the reflection profile. The tomography also enabled the characterization of the Kalahari cover thickness, revealing spatial variations that correlate with mapped outcrops and regional geophysical anomalies (Figure 11).

Receiver function analysis further strengthened our interpretation by offering constraints on Moho depth and crustal layering. The Moho depths inferred from receiver functions align with some of the deepest reflectivity patterns in the seismic profile, providing a more convincing seismic interpretation of crustal-scale structures.

By integrating these datasets, we were able to build a more robust geological model of the Kheis Province, reducing uncertainties inherent in any single method. The combined approach enhances confidence in the interpretation of key structural features and their implications for regional tectonics.

#### 5.2 Evaluation of major tectonic boundaries





Regarding the Dabep Thrust, a lack of clear, continuous reflection(s) or significant velocity contrast at the proposed location of this thrust (Moen 1999) provides little support for the proposition of this being the location of a major tectonic boundary. Similar verdicts have been given by authors of recent geophysical and geological studies of the area (Corner and Durrheim, 2018; Van Niekerk and Beukes, 2019) that challenge the existence of the Dabep Thrust as a first-order crustal-scale boundary. Instead, the structural complexity in this region appears more consistent with a diffuse zone of deformation (Figure 8) rather than a discrete, well-defined thrust fault. However, the absence of the fault's clear manifestation in the seismic data does not conclusively deny its physical existence and could be explained by other attributes such as bad geophone coupling. If the Dabep is indeed present and, in light of the short, discontinuous reflectivity and lack of velocity/Moho offsets, we interpret the Dabep as a candidate minor back-thrust, analogous in style to Thrust A but of lesser displacement, likely active early during

contraction and subsequently abandoned. Our data do not require significant deeper-level extensional reworking along this structure.





The Kalahari Line emerges as a more convincing boundary in our analysis, consistent with its regional magnetic signature. However, its seismic manifestation is complex. Rather than appearing as a tightly bound, well-defined fault plane, the Kalahari Line coincides with a broad zone of seismic transparency in the reflection profile, which does not tessellate well with its narrow magnetic anomaly (Figure 8). Generally, a narrow magnetic anomaly correlates with shallow structures, and the lateral correlation of this feature with high-velocity contrasts in the tomography (Figure 11) suggests that it coherently marks a fundamental boundary at least in the shallow lithosphere. At greater depths, the Kalahari Line has been described as a feature that demarcates a discontinuity in basement depth (Corner and Durrheim, 2018). A consistent interpretation could indicate a steeply dipping structure or a zone of distributed deformation that does not yield strong coherent reflections. It may have functioned as a zone of strain partitioning during oblique convergence, with its steep, reflective-disrupted character reflecting an accommodation of transverse strain complementary to shortening on lower-angle thrusts elsewhere.

In contrast to the Dabep Thrust, the Blackridge Thrust is clearly resolved in the seismic data as a west-dipping reflection with an average dip of ~28° that extends to the surface at a mapped fault location (Figure 7). The thrust underlies a package of chaotic reflections, which likely correspond to imbricated thrust sheets within the Olifantshoek Supergroup. The strong seismic signature of the Blackridge Thrust confirms its role as a significant structural boundary separating the Kheis Province from the Kaapvaal Craton.

Our results provide new seismic evidence supporting the presence of Thrust A, as proposed by Van Niekerk and Beukes (2019). In the reflection profile, Thrust A appears as a discrete, west-verging reflection that truncates underlying reflectors (Figure 8). The geometry of these underlying reflections is consistent with the structural interpretation of a back-thrust within the Kheis Province. Additionally, Thrust A's location coincides with a subtle magnetic anomaly that could be supportive of its identification as a thrust plane.

The Brakbosch-Trooilapspan Shear Zone is difficult to definitively outline, due mainly to a series of near-vertical reflection truncations and zones of seismic transparency (Figure 9). This pattern suggests a network of steep planes that are not detected with the given seismic acquisition. The interpretation posed in Figure 9 that the Trooilapspan Shear Zone intersects units within the Areachap terrane is consistent with interpretations from aeromagnetic data, where it has been mapped as a major transcurrent fault system (Van Niekerk and Beukes, 2019). The localized nature of reflections within this zone indicates strong internal deformation, which may have resulted in significant fabric destruction and attenuation of seismic signals.

While there has been substantial improvement in the imaging quality of the seismic data via reprocessing, there is still a significant lack of clearly defined reflections in the western portion of the GS-02 profile, making it difficult to outline exact terrane boundaries in the region. We cannot therefore make a clear delineation of the eastern boundary of the Areachap province.

### 5.3 New findings from integrated results



The first-break traveltime tomography results have provided a well-resolved estimate of the Kalahari cover thickness along the seismic profile. The results indicate an average sediment thickness of 250 m, with localized variations reaching up to 360 m in the central part of the profile (Figure 11). The thickest deposits are found between the Langeberg and Skurweberg mountains, while areas of zero cover coincide with mapped Proterozoic outcrops. This refined thickness model is consistent with previous borehole and geophysical estimates (Haddon, 2004) but offers a more spatially continuous and detailed characterization along the seismic line.

The reprocessed seismic data provide evidence for a series of plunging fold structures in the western part of the profile, specifically the Orange River Syncline and the Gariep Anticline-Syncline pair (Figure 9). These folds, initially posited from aeromagnetic data analysis (Van Niekerk & Beukes, 2019), are now imaged in the seismic profile as concave-up and concave-down reflectors with localized structural complexity. The results confirm that these structures are related to crustal shortening events associated with regional deformation in the Kheis Province.

The identification of eastward-dipping reflectors near the centre of the profile suggests backthrust features within the Kheis Province, particularly near Thrust A (Figure 8). These structures appear as discrete, west-verging reflectors that are truncated by shallower reflections. The presence of these backthrusts suggests that the Kheis' deformation history involved complex thrust stacking, potentially accommodating variations in crustal shortening across different lithologies.

#### 5.4 Crustal Model

Figure 12 shows a summary of the integrated data at crustal scale (GS-02 and KBF-01 reflection seismic lines, and the 1D crustal models at UPI, SA22 and SA23). Figure 12a shows the RF-Vsapp inversion velocity models superimposed on the composite seismic reflection profile. Figure 12b shows a joint interpretation of both the reflectivity and the major interfaces from the 1D velocity models from the broadband stations. Figure 12c shows a final first-order crustal-scale interpretation of the combined datasets. Here, we describe our inferential process and rationale behind the construction of this interpretation based on the data presented in Figures 8, 9, 10, and 13.

Figure 12: First-order crustal interpretation based on integrated reflection seismic and RF results. From top to bottom, (a) shows the seismic station inversion results overlaid on the reflection seismic section, (b) shows the interpolated horizons of the RF results, and (c) shows the resulting crustal model.

The most robust interpretation in the composite seismic section is the eastern part, where the crust of the Kapavaal craton, including the top supracrustal units, are underthrusting towards the west. The KBF-01 reflection profile is limited to a depth of 20 km, but the RFs provide insight into the crustal stratification, which, together with KBF-01, can be confidently interpolated to the easternmost edge of the deeper GS-02 reflection profile. The supracrustal units can be followed from the top 10-15 km in the east and dipping to depths of 10-35 km in the central part of the profile (Figures 8, 9 and 13). The underlying crustal interfaces follow this general west-dipping trend, but the dip diminishes with depth, and the Moho beneath the Kaapvaal is almost horizontal, suggesting that the crust and particularly the lower crust may have experienced considerable modification syn- and post-collision and has been re-equilibrated or re-worked. The westernmost limit of the underthrusted supracrustal units of the Kaapvaal exhibit a distinct reflectivity package, which curves upwards to the west (Figure 12: right stippled circle; Figure 8). The apparent repetition of this reflectivity pattern appears ~20 km to the west, near CMP 13000 and at a slightly shallower depth. If these fabrics are related, this suggests a crustal-scale normal fault that has potentially offset the westernmost Kaapvaal upper crust, possibly to have occurred after tectonic collision during orogenic relaxation and

extension. A similar structure has been suggested by Stettler et al., (1999), who interpreted this as the Kheis-Kaapvaal suture zone, with no clear account of the offset between the similar reflectivity patterns. The existence of this potential extensional feature is highly speculative, and may well be related to the Kheis-Kaapvaal suture. If the reflectivity pattern belongs to the Kaapvaal, then it is sandwiched between upper and lower Kheis crust, forming a "crocodile" structure, where the Kaapvaal upper crust has indented/buttressed into the mid-crust of the Kheis terrane. Alternatively, if it belongs to the Kheis Province, then there may be plausibility in the assertion that the suture between Kheis and Kaapvaal provinces is represented by the crustal-scale east-dipping lineament. In any case, we attribute the lower crust in the central part of the profile to the Kheis province. Above the mid-lower crust, the reflection data suggest a "saucer"/lens shaped structure, approximately 70 km long and 15-20 km thick, with east-dipping reflectors on its western flank and west-dipping reflectors on its eastern flank, bulging up in the centre of the profile to almost-surface level, where Thrust A is observed on the surface (Figure 8; Can Niekerk and Beukes, 2019). We interpret this to be one unit, possibly of thrusted and deformed Kheis upper crust. However, Stettler et al. (1999) interpreted the Kheis-Kaapvaal suture to continue to near the surface and divide this unit into two, which they justified by a contrast in conductivity and modelled density. While they invoke a model that this east-dipping crustal-scale lineament represents the Kheis-Kaapvaal suture, this model is tectonically difficult to accommodate the underthrusting Kaapvaal units. Instead, we place preference in a model in which the lower Kaapvaal crust underthrusted the Kheis terrane, with the Blackridge Thrust delineating the thrust front, whereas the upper crust detached and indented into the mid- Kheis crust.






Basement is identified in SA22 at a depth of about 5 km, placing it almost colinear with the reflection that we interpret as "Thrust A" from Van Niekerk and Beukes (2019) (Figure 8b). Given that the reflection data comprises the detection of 610 primarily p-waves, the low amplitude of the reflection is not in conflict with a sharp Vs discontinuity. If these two features do indeed represent the same structure, then the thrust fault is accompanied by a sharp increase in the Vs of about 200 m.s<sup>-1</sup>, and could possibly form the eastern limb of a shallower basin whose western limb comprises the folding features identified in Figure 9b. However, this interpretation does not explain the Kalahari Line and would need more evidence than the coinciding 615 SA22 Vs discontinuity with the subtle truncating reflection. The disparity between the RF Moho at station SA22 (~32.5 km) and the deepest reflectivity in the central part of the profile (~38-40 km at ~110 km profile length) can be explained by the offline position of station SA22. The RF result is a representation of the crustal structure ~20 km to the north and ~20 km to the west of the projected location, suggesting thinning of the crust to the northwest. The RF waveforms from events with southerly backazimuth crossing the Moho closest to the seismic line do show signals at greater delay time of ~5 sec compared to all other directions, which could indicate a step or gradient in Moho depth from the location of SA22 and the seismic line.

The western part of the profile is less constrained, but we interpret the majority of the basement to belong to the Kheis terrane with a 40-45 km thick crust, based on both reflection seismics and receiver functions, which is roughly constituted of 50% upper and 50% lower crust. However, the upper and middle crust hosts some strong complexity. As discussed in the Results, one striking feature is the localized high-reflectivity pattern located at ~50-60 km offset and 15-20 km depth. The feature resembles a narrow rift basin; however, cross-cutting reflectors may also suggest additional intrusive magmatism. The area of high, layered reflectivity is overlain by a markedly transparent unit. Deeply buried basins are not unknown, even at these depths, however, they are usually buried beneath younger sedimentary basin, which is not suggested by the reflection data. Transparent areas in reflection seismics represent typically homogenous zones such as massive intrusions or partial melt regions. The transparent zone is relatively well-defined and contrasts with surrounding, more reflective regions. We speculate that this zone is therefore attributed to a plutonic body, possibly granitic or gabbroic, with relatively uniform composition and few internal impedance contrasts, leading to low reflectivity. Further details, including the origin and relationships of this structure to the surrounding rock requires more complimentary analyses to provide a more confident interpretation.

The western edge of the profile exhibits reflectors dipping to the west from the surface, which we interpret as likely connected to the Brackbosch-Trooilapspan Shear Zone (Figure 9, 12). In this area, rocks from the Areachap island arch appears on the surface geology and the expected cross-section of the crust. We follow the interpretation by Stettler et al. (1998), which posits this as the first occurrence of the island arc, whereas the mid and lower crust still belongs to the Kheis basement.

#### **6 Conclusions**


- In this study, we presented a comprehensive reassessment of the crustal-scale structures and tectonic domains of the Kheis Province using reprocessed seismic reflection data, seismic tomography, and receiver function analysis. By integrating these methods, we provide new insights into key tectonic boundaries, crustal architecture, and the evolution of the region, with the goal of shedding light on features that will provide a foundation for future tectonic studies in the area.
- The reprocessing of the GS-02 seismic reflection profile, particularly through prestack Kirchhoff migration, significantly improved imaging of crustal structures. Key structural elements such as thrust faults, fold geometries, and deep-seated reflective packages were resolved with greater clarity than in previous interpretations. Despite these improvements, 2D seismic imaging remains limited in resolving 3D structures, emphasizing the need for complementary geophysical data.
- Refined interpretations of major tectonic boundaries can be summarized as follows. The Dabep Thrust lacks a clear seismic expression, supporting recent studies that question its role as a major tectonic boundary. The Kalahari Line emerges as a significant structural boundary, though its seismic expression is complex, likely due to steep faulting or distributed deformation. The Blackridge Thrust is well-defined in the seismic profile, clearly coincident with the thrust front that separates the Kheis Province from the Kaapvaal Craton. New evidence supports the presence of Thrust A, located near the midpoint of the profile section that spans the aeromagnetic fabric of the Kheis Province. The Brakbosch-Trooilapspan Shear Zone is observed as a zone of seismic transparency and disrupted reflections, consistent with transcurrent faulting. The identification

of anticlines and synclines, including the Gariep and Orange River fold structures, support recent literature positing their presence based on surface geology and aeromagnetic data.

In our near surface studies, first-break tomography results provide a refined thickness model for the Kalahari cover, averaging 250 m. At greater depths, receiver function analysis confirms Moho depths varying from 32 km (SA22) to 46 km (SA23), aligning with the deepest reflectors in the seismic profile. Our preferred model of the crustal structure involves westward underthrusting of the Kapvaal Craton under the Kheis terrane with partial imbrication of the Kapvaal crust into the Kheis crust at mid-crustal levels, forming a classic "crocodile" structure, frequently observed in continent-continent collision zones (e.g. Meissner 1989, Meissner et al., 1991). While unexpected, our tectonic model of a thinner crust near the Kaapvaal-Kheis transition agrees with other published results. We also emphasize the enigmatic presence of the ~15-km-deep basin structure in the western part of the profile with a lateral span of 12 km.

While the structural complexity of the Kheis Province means that a definitive, coherent tectonic model remains elusive, this study provides a valuable stepping stone toward a more comprehensive understanding of its geological and tectonic evolution.

## **Author Contribution**

MW and MM were jointly involved in the conceptualization and data curation for the project. All authors were involved in the formal analysis, methodology and interpretation of the various datasets. MW conducted writing of the original draft, and all authors were involved in the review and editing of the manuscript.

#### 675 Competing Interests

The authors declare that they have no conflict of interest

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

# Appendix A

810 Figure A1: Full plot of GS-02 profile, with (a) showing seismic amplitudes, and (b) showing amplitude envelope.

Figure A2: Backazimuth plots of RFs at stations UPI (a), SA22 (b) and SA23 (c). The RFs were stacked in bins of four quadrants pointing to the north (-45 to 45 degrees with respect to north), south (135 to 225 degrees), east (45 to 135 degrees) and west (225 to 315 degrees). Italic numbers with a yellow background indicate the number of events in each stack. Stacks of more RFs are naturally more robust representations than stacks with fewer events. Note that the southern quadrant is represented twice at 180 degrees.