# Peer review of "Crustal-scale structures and tectonic domains of the Kheis Tectonic Province in South Africa from multimethod seismic analysis"

_EGUsphere, 2025_

## Referee Comment (RC2)

[revised manuscript text omitted]
. West of this package, three near-vertical zones of distinct reflectivity are seen, which are characterized by sparse reflections that are truncated at the zone boundaries. These boundaries are thus interpreted as faults belonging to the Brakbosch-Trooilapspan shear zone (Figure 2). Their surface locations coincide with the respective boundaries as interpreted by Corner and Durrheim (2018) and Van Niekerk and Beukes (2019). Fabric destruction along the steep fault zone may explain the scarcity of continuous reflections within the region. The lack of notable reflections west of CMP 17500, sparing a few that are discontinuous and dip steeply to the west, are likely explained by steep bedding in the Areachap deposits.

[revised manuscript text omitted]

---

## Author Response (AR1)

Title: Crustal-scale structures and tectonic domains of the Kheis Tectonic Province in South

Africa from multimethod seismic analysis

Author(s): Michael Westgate et al. MS No.: egusphere-2025-1844 MS type: Research article

Special issue: Seismic imaging from the lithosphere to the near surface

**RESPONSE TO REVIEWER AND EDITOR COMMENTS**

(All author responses to comments are formatted in bold blue)

**Reviewer 1**

This paper presents a fresh perspective on the Kheis tectonic province (South Africa) and adjacent terranes, based on an integrated analysis of previously published seismic data. Although the datasets are not new, their combined use is original and further enhanced by the reprocessing of the GS-02 deep reflection seismic line, traveltime tomography of first arrivals, and teleseismic receiver function analysis. The authors provide a clear overview of the key geological elements and existing tectonic models, effectively highlighting the role of geophysical datasets in shaping these interpretations. This serves as a valuable foundation and contextual summary of previous work in the region. The methodology is clearly described, and the interpretation carefully outlines the principal reflections observed in the 2D seismic profiles. The discussion concisely summarizes the main results, distinguishing features supported by the data from those that are not, and highlighting new structures identified in this study. The manuscript is well-written, well-organized, and clear. The figures are of high quality and effectively illustrate the key data and results. I have only minor comments and suggestions for improvement.

**Comparison of the 1998 vs reprocessed seismic section:**

Figure 6 is somewhat difficult to evaluate, as no processing details are provided for the 1998 section. There is a general concern that the comparison may not be equivalent. It is unlikely that PSTM was applied to the 1998 data, and improved reflection focusing in the reprocessed section could reasonably be attributed, at least in part, to the application of PSTM. However, the reprocessed data includes semblance filtering, which does not appear to have been applied to the 1998 section. Without such filtering, the earlier data would naturally appear of lower quality, especially at the scale shown. I suggest including a subfigure of the reprocessed data prior to semblance filtering to allow readers to better assess the relative contributions of PSTM (and processing steps preceding PSTM) and semblance filtering to the observed improvements.

As the reviewer has pointed out, there is unfortunately insufficient information available about the original processing flow to make a direct comparison throughout the flow. However, regarding the semblance filtering contribution, we have adopted the suggestion to include a subfigure that shows the reprocessed data prior to semblance filtering as part of Figure 6, as well as some discussive text (Line 305-308).

Line 82: "Additionally, A reappraisal...". Replace with: Additionally, a reappraisal...

Corrected.

Line 136: ... supracrustals: replace with supracrustal rocks.

**Corrected.**

Line 152: can you please specify the metamorphic grade?

**Updated.**

Line 206: Please define RF.

**Updated.**

Lines 257 and 261: There is no need to define 'RF' as it is defined at line 206.

**Corrected.**

Line 296: Can you please provide details on the time-to-depth conversion function?

(New Lines 291-294) A sentence has been added to detail how the time-do-depth conversion was added.

Line 396: Please define HVLC.

**Updated.**

Line 452: "...with prestack Kirchhoff migration yielding the most substantial improvements." As discussed above, it would be valuable to distinguish the contribution of PSTM from that of semblance filtering.

**Updated as above.**

Line 479: "Similar sentiments...". Not sure "sentiments" is the most appropriate word.

**Updated.**

Line 632: "MW and MM wee..." were?

**Corrected.**

Figure 2d. What is the unit in yellow (no yellow box in the legend)?

**Updated.**

Figure 7b: What are the units in light orange and pink?

**Updated.**

Figure 9b: The presence of the Trooilapspan Shear, the Brakbosch Fault, and the fault near CDP 17,500 is not clearly supported by the seismic data. On what basis are these structures extended to 30 km depth in Figure 9b? What evidence justifies depicting them as vertical features to that depth?

We have revised the interpretation in the western section of the profile. The vertical Trooilapspan and Brackbosch structures altered to have a westward dip, and are in more agreement with the surficial interpretation. (See Figures 9, 12, and Lines 382-398; 526-537).

Figure 10: What are the lines in the subfigures (full and dashed)?

**Updated figure caption.**

**General comments:**

The paper by Westgate and colleagues integrates reprocessed deep crustal reflection seismic data with receiver functions to investigate the crustal architecture and (proposed) major structures along a transect from the southwestern Kaapvaal Province through the Kheis Province into the Namaqua-Natal Metamorphic Province. The authors also use first arrival tomography to map cover thickness/depth to basement along the transect. The reprocessing results of the reflection data do represent a substantial enhancement of the legacy seismic data set and provide new constraints on the geology of the area, which is subject to considerable debate. Therefore, the work presented is an engaging topic and of interest to the readership of Solid Earth. The paper is in general well written. Although the illustrations are overall of good quality, there are some aspects that in my opinion require improvement and/or further clarification/explanation to make the manuscript more accessible to the reader. I have also some concerns and questions regarding aspects of the interpretation as laid out below.

I provide some comments and suggestions for improvements that would further strengthen the manuscript below and the annotated file attached. I have no doubt my concerns can be addressed by the authors and am happy to recommend publication after minor to moderate revision.

**Specific comments:**

Interpretation of steep structures: The authors interpret a range of steep structures in the reflection seismic data. For some of them, I have problems to reconcile their position with the map provided in figure 2. For example, the Brakbosch Fault does not reach the seismic line according to the map, and the authors refer to the Brakbosch-Trooilapspan shear zone as one coherent structure (line 154). For others (Kalahari Line, Trooilapspan Shear), the map expression is not that of a subvertical fault. Lastly, none of these faults seems to have any effect on the crustal model the authors propose in figure 12 c – is that realistic for such major structures? These are important points that need to be addressed and my main concerns with the manuscript.

We have revised the interpretation to conform with both reveiwers' suggestions and in a way that is more representative of the seismic data. We have reinterpreted the Trooilapspan-Brackbosch Shear Zone to be possibly coincident with dipping reflectors in the western edge of the seismic data, and not as sub-vertical. (See Figures 9, 12, and Lines 382-398; 526-537). While the Kalahari line has not been revised, we don't associate it as the direct cause of the transparent zone in the seismic data, but rather as related to it. We have also adapted the text to make that clear (Lines 357, 507-508).

Interpretation: Some of the interpretations are also not very obvious in the data in my opinion. For example, the Brakbosch Fault seems to cross coherent reflectors at c. 17 km depth, CMP ~16 450. On the other hand, the (moderately west dipping) boundary of the Areachap Terrane as interpreted in Figure 12c seems to project to the position of the Brakbosch Fault – so why does the Brakbosch Fault not represent this terrane boundary? And what is this terrane boundary

called if not the Brakbosch Fault? Where is this Areachap Terrane boundary mapped in the detailed interpretation figure 9?

We have revised the overall interpretation of the western part of GS-02, along with Figures 9 and 13, and adopted a more conservative approach to the interpretation that is now more coherent. Specifically, regarding the boundary of the Areachap domain, the reviewer points out that the Brackbosch-Trooilapspan shear zone is one continuous boundary. This is the case according to conventional sources of literature (such as Moen, 1999), however, in recent literature (Van Niekerk and Beukes, 2019), these structures are seen as not necessarily continuous. We have added text at the end of section 4.1 (Lines 387-393) to clarify and elaborate on this point. We have also updated Figure s 9 and 13 that provide a more clear, broad eastern boundary for the Areachap province. We have chosen to refrain from a defining a single boundary for the province as there is no strong evidence for it in the seismic data, but have placed it to the west of the shear zone.

Buried valley vs rift basin: One more reflective area in the western part of the profile is first interpreted as a "buried valley" (line 376), and then later in the discussion and conclusion as a "resembling a narrow rift basin". This is inconsistent and confusing, and while I don't really agree with either option a "buried valley" seems more difficult to reconcile with that data, given that the feature in question has a thickness of 8 km or so. In terms of the rift basin, how do the authors envisage the structural context (i.e. bounding faults etc).

We agree with that the rift basin theory is more plausible and have updated the text to both elaborate on the anomalous nature of the feature (Lines 376-381), and to suggest the structural context of the feature as a rift basin, per the reviewer's recommendation (See Figure 9).

Integration of reflection seismic with receiver functions: I do commend the authors for their approach to integrate both datasets as much as possible, which improves the overall interpretation. However, that can be a challenge in cases where the results are somewhat contradicting, and a balance needs to be struck as how to weigh the different datasets. In my opinion, the information of station SA22 is overrepresented in the final interpretation (Figure 12). SA22 is some 30 km away from the seismic profile. It suggests a very shallow Moho at ~32 km, which the authors project onto the seismic transect in their final interpretation. At the position of the profile, the reflection seismic data suggest a deeper Moho around or just over 40 km, and the authors map corresponding reflections marking the lowermost crust in Figure 12b. This depth is much more aligned with the overall crustal thickness of the terrane clearly imaged to the west and does not require a pronounced step in the Moho that is seemingly not related to a major structure and/or difference in the lower crustal geology (e.g. a different terrane). I therefore suggest considering giving preference to the Moho as imaged in the reflection seismic, avoiding the uncertainty related to the 30 km projection. I also note that prior studies on SA22 have suggested Moho depth between 30 and 48 km (lines 205 – 209).

We have included backazimuth distribution plots of earthquakes recorded by all three stations (Figure 10) and updated our discussion around the receiver functions to consider the distribution of incoming signals, as well as to reconcile the data with the reflection section (See Lines 424-425, 435-437, 445-447, 602-606). As a result, we have also updated the position of the Moho in Figure 12 to be deeper as per the reviewer's comment.

**Figures:**

Minor comments on the figures are provided in the annotated pdf, but I would like to raise two points here:

(1) To allow the reader to assess the reflection data at crustal scale I'd like to encourage the authors to show the full depth extend of the reflection data for the detailed interpretation figures (7 to 9). The Moho topography and the character of the lower crust are both quite relevant when interpretating terrane boundaries and major structures and are difficult to assess in the current form/depth extent.

We appreciate the reviewer's concern and agree that clarity in imaging crustal-scale features is essential when discussing structures such as the Moho. However, we chose to limit the depth extent of the zoomed-in seismic figures, and we would like to keep it the way it is, for several reasons: (1) We aim in these figures to highlight key identified reflection signals. Below the zoom limits, the signal quality diminishes significantly (see Figure 5), and no coherent reflectors are present, providing no detailed additional interpretive information and would visually dilute the focal structural details that are being highlighted. (2) Our interpretation of the Moho is not based on discrete reflectors within these zoomed-in sections but rather on regional changes in seismic facies observed in the broader profile (Figures 5, 12, and A1), as well as receiver function analysis. (3) To provide full crustal context, we believe the zoomed-out figure later in the manuscript (Figure 12) and the high-resolution, uninterpreted version of the entire seismic line in the appendix (Appendix Figure A1) allow the readers to assess the full depth and our interpretations independently.

(2) in the overview maps (figure 2) and in the detailed interpretation (figures 7 to 9) CMP is used for reference, location and for descriptions in the text. In figure 12 the authors use kilometres along the line instead, also for the description (e.g. line 583), which makes navigation and cross correlation between figures difficult. Please use CMP throughout for consistency.

We have updated the manuscript to include CMP numbers throughout the text, and in Figure 12, in addition to kilometres.

Two reviewers have now provided substantive comments and both agree that the manuscript would make a suitable contribution to this special volume of Solid Earth. Both raise important concerns however, and the authors should address these concerns as the changes suggested will greatly improve the readability of the manuscript. In particular, more clearly document the improvement achieved by reprocessing and suggest what type of data (e.g., bandwidth, sampling rate) could be similarly improved. The questions as to which Moho to adopt and where to interpret faults are controversial in general and do deserve more discussion and clarification here. I faced very similar concerns in my recently accepted contribution to this volume, and the authors may wish to consider the various arguments in the paper, reviews and replies as they revise their manuscript. Personally, I value a good receiver function over reflection signatures, but would use multi-azimuthal receiver functions to access whether 30 or 45 km is more probable or if a step in the Moho does exist near station SA022. Interpreted fault locations at depth should be consistent with the fault type and its mapped trace and sense of offset at the surface.

We have addressed the respective concerns and edits by each reviewer independently and agree that the suggestions have greatly improved the strength and readability of the manuscript. Please see our individual and detailed responses to the reviewer comments. As to the main points raised: The comparison between the legacy data and the reprocessed data has been refined to include a non-filtered comparison, as per reviewer 1's suggestion. Regarding the discussion around the Moho depth near the SA22 station and its reconcilability to the reflection data, and as noted with reviewer 2, we have updated the manuscript to included azimuthal information of the earthquakes recorded by the stations. This allowed us to discuss the results further and update our Moho depth to be more consistent and realistic. Finally, regarding the interpretation of vertical structures that was raised by both reviewers, we have significantly altered our interpretation of the western portion of the GS-02 profile to conform more with the other datasets and the overall tectonic model, an alteration that we agree has more credibility.

---

## Author Response (AR2)

Title: Crustal-scale structures and tectonic domains of the Kheis Tectonic Province in South

Africa from multimethod seismic analysis

Author(s): Michael Westgate et al. MS No.: egusphere-2025-1844 MS type: Research article

Special issue: Seismic imaging from the lithosphere to the near surface

**RESPONSE TO EDITOR COMMENTS ACCOMPANYING MINOR REVISION**

(All author responses to comments are formatted in bold blue)

**Editor**

Most of the reviewers' concerns were adequately addressed, but the multi-azimuth RFs continue to be under-utilized. Station SA22 lies 30 km north of the profile so diving rays arriving from the south will cross the Moho about 15-20 km closer to the profile than those arriving from the north. Key question here is whether the RF 'spikes' at 39 and 46 km each come predominantly from N or S and thus define a N-S slope to the Moho? Station SA23 lies on the profile so, similarly, do rays from the E and W cross the Moho at 33 and 38 (46??) km, respectively?

We have now added plots in the appendix of RF stacks that are binned in the four quadrants (north, south, east, west) and used this as an additional basis for discussion. The outcomes show that the RFs at SA22 exhibit a different waveform to the south, whereas at UPI and SA23, they are more uniform with respect to backazimuth. However, the number of RF in each bin varies, and therefore also the data quality. The bins with few events will not have as robust RF stacks as the bins with more events, which makes a direct comparison a little difficult, but it might still be illustrative for the reader. The updated discussion points in the text are found at lines 406-408; 427-428; 438-442; 450-455; 618-621. Figure 10 has also been revised to show rose diagrams plotted in degrees, not radians.

Consider whether the Dabep (like A, line 553) Thrust might represent an early, minor back-thrust to the main collision, later abandoned to deeper level extension as you currently suggest. Similarly, the Kalahari Line (line 506-514) might represent strain partitioning during convergence, with this vertical zone absorbing transverse strains during regional oblique convergence.

We agree that, given the belt-parallel kinematics and the presence of back-thrusting (e.g., Thrust A), an interpretation of the Dabep as an early, minor back-thrust is plausible. We have added some appropriate speculative text to the discussion (Lines 505-509).

Regarding the Kalahari Line, with its steep geometry and zone of seismic transparency, it is well-suited to have acted as a strain-partitioning structure during oblique convergence. Its geometry in the data (broad, vertically extensive zone of disrupted reflectivity) is consistent with an accommodation zone that absorbed transverse (strike-slip) strains.

We have incorporated this interpretation into the text (Lines 518-520) to highlight the possibility that the Kalahari Line played a role in partitioning deformation during

**convergence.**

line 50 Add Juhlin et al. this volume ref? (my role trying integrate papers within this special volume)

line 186 Transvaal and Olifantshoek Supergroup units crop out... line 499 (Moen, 1999)

The respective additions/corrections have been implemented.